# Comparison of Pretraining Models and Strategies for Health-Related Social Media Text Classification

**DOI:** 10.3390/healthcare10081478

**Published:** 2022-08-05

**Authors:** Yuting Guo, Yao Ge, Yuan-Chi Yang, Mohammed Ali Al-Garadi, Abeed Sarker

**Affiliations:** 1Department of Biomedical Informatics, Emory University, Atlanta, GA 30322, USA; 2Department of Biomedical Informatics, Vanderbilt University Medical Center, Nashville, TN 37240, USA

**Keywords:** machine learning, social media, text classification

## Abstract

Pretrained contextual language models proposed in the recent past have been reported to achieve state-of-the-art performances in many natural language processing (NLP) tasks, including those involving health-related social media data. We sought to evaluate the effectiveness of different pretrained transformer-based models for social media-based health-related text classification tasks. An additional objective was to explore and propose effective pretraining strategies to improve machine learning performance on such datasets and tasks. We benchmarked six transformer-based models that were pretrained with texts from different domains and sources—BERT, RoBERTa, BERTweet, TwitterBERT, BioClinical_BERT, and BioBERT—on 22 social media-based health-related text classification tasks. For the top-performing models, we explored the possibility of further boosting performance by comparing several pretraining strategies: domain-adaptive pretraining (DAPT), source-adaptive pretraining (SAPT), and a novel approach called topic specific pretraining (TSPT). We also attempted to interpret the impacts of distinct pretraining strategies by visualizing document-level embeddings at different stages of the training process. RoBERTa outperformed BERTweet on most tasks, and better than others. BERT, TwitterBERT, BioClinical_BERT and BioBERT consistently underperformed. For pretraining strategies, SAPT performed better or comparable to the off-the-shelf models, and significantly outperformed DAPT. SAPT + TSPT showed consistently high performance, with statistically significant improvement in three tasks. Our findings demonstrate that RoBERTa and BERTweet are excellent off-the-shelf models for health-related social media text classification, and extended pretraining using SAPT and TSPT can further improve performance.

## 1. Introduction

Supervised text classification is perhaps the most fundamental machine learning task in natural language processing (NLP), and it has been employed extensively to design data-centric solutions to research problems within the broader biomedical domain. Formally, this task involves the training of machine learning models using a set of text (often referred to as records or documents in early research) and label (also referred to as class or category) pairs, where the number of labels is finite, and then employing the trained model to automatically predict the labels for previously-unseen texts [1]. Compared to supervised classification of structured data, text classification typically poses additional challenges due to the presence of large feature spaces (i.e., high dimensionality of feature space) [2,3] and feature sparsity [4,5]. Approaches such as support vector machines (SVMs) [6], Random forests [7], and logistic regression [8] had produced state-of-the-art (SOTA) classification performances for many tasks over the years due to their abilities to handle large feature sets consisting of bag-of-words or n-grams. These traditional approaches typically relied on feature engineering methods to generate salient features from texts, and improve performances particularly by addressing the feature sparsity problem. Text classification tasks within the medical domain primarily benefited from domain-specific features, often generated via the utilization of knowledge sources such as the unified medical language system (UMLS) [9]. With the emergence of methods for generating effective numeric representations of texts or word embeddings (dense vectors), coupled with advances in computational capabilities, deep neural network based approaches became dominant in this space, obtaining SOTA performances in many text classification tasks [10,11]. Such approaches use dense vector representations, and generally require large volumes of annotated data. Word embedding generation approaches such as Word2Vec [12] and GLoVe [13] are capable of effectively capturing semantic representations of words/phrases (i.e., text fragments with similar meanings appear close together in vector space), which n-gram based approaches were not capable of. However, these context-free embedding generation approaches do not provide any mechanism for disambiguating homonyms (e.g., the term ‘bank’ in ‘river bank’ and ‘bank cheque’ would have the same vector representation). This limitation was overcome relatively recently via the proposal of transformer-based models that are capable of capturing contextual vector representations for texts.

Pretrained transformer-based models such as bidirectional encoder representations from transformers (BERT) [14] and RoBERTa [15] have achieved SOTA results in most domain-independent NLP tasks (i.e., tasks involving generic texts), often with substantial performance increases over past SOTA approaches. Recent research efforts attempted to boost the performances of pretrained transformer-based models on domain-specific tasks by domain-adaptative pretraining (DAPT), which involves further training of a generic pretrained model such as BERT on domain-specific data. For example, Lee et al. [16] proposed BioBERT by pretraining BERT on a large biomedical corpus of PubMed abstracts, and demonstrated that it outperforms BERT on three representative biomedical text mining tasks. Alsentzer et al. [17] attempted to adapt pretrained models for clinical text by training BioBERT on clinical notes, resulting in the creation of BioClinical_BERT [18]. Gururangan et al. [19] illustrated the usefulness of DAPT by continuing training of pretrained models on domain-specific data from four different domains (biomedical and computer science publications, news, and reviews). However, some studies, including our own pilot, demonstrated that DAPT is not guaranteed to achieve SOTA results for health-related NLP tasks involving social media data [20,21]. Similar findings were observed even in studies that achieved SOTA results by DAPT on multiple tasks. For example, the experimental results in Gururangan et al. [19] show that DAPT did not increase performance on one task with news domain data. Alsentzer et al. [17] shows that the model by training BioBERT on clinical notes only outperformed BioBERT on two out of five clinical NLP tasks. To address such performance issues, several studies have experimented by continuing pretraining on social media data (we refer to it as source-adaptive pretraining; SAPT), and demonstrated their superior performance on social media specific NLP tasks [22,23].

Data from social media, often referred to as consumer-/patient-generated data, are increasingly being utilized for health-related research [24,25,26]. A large population of patients are actively involved in sharing health related information in social networks and online health forums [27]. The healthcare information discussed on social media is useful for health and disease surveillance, such as tracking infectious disease outbreaks [28,29,30], monitoring signals associated with mental health problems such as sleep problems, depression, post-traumatic stress disorder (PTSD), and suicide [31], to name a few. Pharmacovigilance has also been a major area of social media based health research [32,33]. Several studies have shown that traditional data sources from the medical systems may be inadequate for post-marketing adverse drug reaction (ADR) monitoring [34,35,36]. For example, the case reports may not be timely and accurate because of under-reporting, over-reporting of known ADRs, incomplete data, and duplicated reporting–a gap that social media data can fill. Broadly speaking, publicly available social media data can serve as an additional data source that allows biomedical and public health researchers to directly study human behavior in a large-scale quantitative manner.

Social media has several attractive characteristics—large volumes of data are available, are generated directly from large segments of the population, can be captured in close to real-time, and can be obtained with little to no cost, to name a few. However, from the perspective of NLP and machine learning, social media presents unique challenges due to the presence of misspellings, noise, and colloquial expressions. NLP of health-related text is itself more challenging compared to NLP of generic text [37,38], and the characteristics of social media data further exacerbate the challenges. Typically, NLP methods developed for generic text underperform when applied to health-related texts from social media. For example, for the task of adverse drug event classification, the same SVM model with identical feature generation methods was shown to exhibit significant performance differences between data from medical literature and social media ($F_{1}$-score dropped from 0.812 to 0.597) [39].

The emergence of transformer-based models and pretraining has thus opened up new opportunities for social media-based health NLP research. However, although recent studies have demonstrated the utility of these emergent models on social media-based datasets, there is a paucity of research available that (i) enables the direct comparison of distinct pretrained models on a large number of social media-based health-related datasets, or (ii) provides guidelines about strategies for improving machine learning performance on such specialized datasets. Pretraining language models is a resource-intensive task, and it is often impossible for health informatics researchers to conduct extensive pretraining or compare multiple pretrained models. In this paper, we investigate the influence of pretraining strategies on performance for health-related text classification tasks involving social media data. In addition, since health-related NLP tasks generally focus on specific topics, we explore a new pretraining strategy—using topic-specific data for extended pretraining (we refer to this as topic-specific pretraining; TSPT)—and compare it with SAPT and DAPT for health-related social media text classification. TSPT can be viewed as a further specialization of DAPT or SAPT, where additional pretraining is performed using data related to the topic only, regardless of the source. Although SAPT could be considered to be DAPT if we consider consumer-generated data to be a different domain compared to biological data, we chose to use the separate terms to disambiguate between non-health-related data from social media (which is not medical domain specific but is source specific) and health-related data from outside social media (which is medical domain specific but not source specific).

A summary of the specific contributions of this paper are as follows:We compare the performances of six models pretrained with texts from different domains and sourcesBERT [14] and RoBERTa [15] (generic text), BERTweet [22], and Twitter BERT (social media text, specifically Twitter) [20], BioClinical_BERT [18] (clinical text), and BioBERT [16] (biomedical literature text)—on 22 social media-based health-related text classification tasks.We perform TSPT using the masked language model (MLM) [40], and assess its impact on classification performance compared to other pretraining strategies for three tasks.We conduct an analysis of document-level embeddings at distinct stages of processing, namely pretraining and fine-tuning, to study how the embeddings are shifted by DAPT, SAPT, and TSPT.We summarize effective strategies to serve as guidance for future research in this space.

## 2. Methods

We used 22 health-related social media text classification tasks for comparing pretrained models. Manually annotated data for all these tasks were either publicly available or had been made available through shared tasks. The tasks covered diverse topics including, but not limited to, adverse drug reactions (ADRs) [39], cohort identification for breast cancer [41], non-medical prescription medication use (NPMU) [42], informative COVID-19 content detection [43], medication consumption [44], pregnancy outcome detection [45], symptom classification [46], suicidal ideation detection [47], identification of drug addiction and recovery intervention [48], signs of pathological gambling and self-harm detection [49], and sentiment analysis and factuality classification in e-health forums [50]. Table 1 presents the details/sources for the classification tasks, the evaluation metric for each task, training and test set sizes, the number of classes, and the inter-annotator agreement (IAA) for each dataset, if available. Eleven tasks involved binary classification, eight involved three-class classification, and one involved four-, five- or six-class classification each. The datasets combined included a total of 126,184 manually-annotated instances, with 98,161 (78%) instances for training and 28,023 (22%) for evaluation. The datasets involved data from multiple social media platforms—11 from Twitter, 6 from MedHelp (https://www.medhelp.org/, accessed on 4 July 2022), 4 from Reddit, and 1 from WebMD (https://www.webmd.com/, accessed on 4 July 2022). We reported the IAA scores from the original papers to inform the readers about the IAA scores for each dataset. For evaluation, we attempted to use the same metrics as the original papers or as defined in the shared tasks.

### 2.1. Evaluation

All system configurations were evaluated against each other based on the metrics shown in Table 1.

### 2.2. Data Preprocessing

To reduce the noise in the Twitter data, we used the open source tool preprocess-twitter for data preprocessing [51]. The preprocessing includes lowercasing, normalization of numbers, usernames, urls, hashtags and text smileys, and adding extra marks for capital words, hashtags and repeated letters. Web content from OpenWebText and research articles from PubMed were chunked into sentences and then preprocessed in the same manner.

### 2.3. Model Architectures

The model architectures for the masked language model (MLM) and classification are shown in Figure 1. MLM is an unsupervised task in which some of the tokens in a text sequence are randomly masked in the input and the objective of the model is to predict the masked text segments. In Figure 1a, the input $\left\{t_{1},\ldots ,t_{n}\right\}$ denotes a text sequence with some tokens masked. The encoder embeds the text sequence as an embedding matrix consisting of token embeddings $\left\{e_{t_{1}},\ldots ,e_{t_{n}}\right\}$. The embeddings of the masked tokens are fed into a shared linear fully-connected layer, and a Softmax layer to predict the masked token. For each masked token, the output is a probability vector that has the same size as the vocabulary. During classification, the individual token embeddings are combined into a document embedding ($e_{d}$) that represents the full instance of text sequence to be classified by average pooling. This document embedding is then fed into a linear fully-connected layer and a Softmax layer to predict the class of the instance.

For extended pretraining, we initialized our MLM models from RoBERTa_Base and BERTweet, respectively, and performed the pretraining on the off-topic, topic-specific and generic pretraining sets we curated. We chose RoBERTa_Base and BERTweet as the initial models for these experiments because they outperformed the other models in our initial benchmarking experiments over the 22 datasets (see Section 3). The generic pretraining was only required to be conducted once for all three tasks, but the topic-specific and off-topic pretraining were distinct for each task. After pretraining, we fine-tuned each model on the target classification task, where the encoder of the classification model was the same encoder of the MLM model. Source code for our model and data preprocessing is available under the Github repository: https://github.com/yguo0102/transformer_dapt_sapt_tapt, accessed on 4 July 2022.

#### 2.3.1. Statistical Significance

In order to better compare the performance of different models, we estimated the 95% confidence intervals for the test score of each model and for the performance difference between the models using a bootstrap resampling method [52]. Specifically, the 95% confidence intervals for the test scores are computed as follows: (i) we randomly chose *K* samples from the test set with replacement and computed the test score of the selected samples; (ii) we repeated the previous step *k* times and to get *k* scores; (iii) we sorted the *k* scores and estimated the 95% confidence interval by dropping the top 2.5% scores and the bottom 2.5% scores. Similarly, when estimating the 95% confidence interval for the performance difference between two models *A* and *B*, we first randomly chose *K* samples from the test set with replacement and computed the difference in test scores ($s_{A}-s_{B}$), where $s_{A}$ and $s_{B}$ are the test scores of the models *A* and *B* on the selected samples. The following steps were the same as the steps (ii) and (iii) as described above. If the 95% confidence interval did not contain zero (i.e., no difference in the test scores), the performances of the models *A* and *B* were considered to be statistically significant. In our experiments, we set *K* to be equal to the size of the test set and set *k* as 1000 for each task.

#### 2.3.2. Document Embedding Transfer Evaluation

Past studies have shown that pretrained transformer-based models can generate embedding vectors that might capture syntactic and semantic information of texts [53,54,55]. Inspired by these works, we attempted to study the effectiveness of SAPT and TSPT by exploring the change in document embeddings following these two pretraining strategies. For each topic, we measured the cosine similarities between the document embeddings of the instances in the training set (*D*) and analyzed the change of document embeddings before and after pretraining. For each document $d_{i}\in D$, there were three document embeddings generated by the following models:$r_{i}$: Default encoder without any modification$p_{i}$: Encoder after pretraining$q_{i}$: Encoder after pretraining and fine-tuning

As described in the previous subsection, both the MLM and classification model architecture contain an encoder, and all the models contained an encoder of the same architecture. The encoder converted each document into an n×m embedding matrix, where *n* is the maximum sequence size and *m* is the dimension of the token embeddings. For each topic, we computed the cosine similarity of the embedding pairs ($r_{i}$, $p_{i}$) and ($p_{i}$, $q_{i}$) in the training set and then analyzed the distribution of cosine similarities by histogram visualization. Our intuition was that effective pretraining strategies would be reflected by observable shifts in the document embeddings, which would be discernible from the histograms. Significant shifts in the document embeddings before and after pretraining would suggest that the models can learn new information from the pretraining data, which can cause performance changes the downstream tasks. Otherwise, further pretraining would be unlikely to affect the performance on the downstream tasks.

### 2.4. Experiments

#### 2.4.1. Data Collection and Preparation

To compare DAPT, SAPT, and TSPT, we required unlabeled data from (i) different sources and (ii) different domains, and (iii) specific to targeted topics. We first collected data from three sources—Twitter (social media; source-specific), PubMed abstracts and full-text articles (medical domain; domain-specific), and OpenWebText (generic/domain independent). For the Twitter and PubMed data, we created additional subsets for TSPT by applying hand-crafted filters. These hand-crafted filters were lists of keywords and/or expressions identified by medical domain experts relevant to each task. The specific filters are described later. Since the process of pretraining is computationally intensive and time consuming, to reduce the time and environmental cost of our experiments, we specifically focused on 3 tasks for extended comparative analysis instead of all 22 tasks—breast cancer, NPMU, and informative COVID-19 tweet classification. We chose the three tasks because it is relatively easier to collect the pretraining data for these three tasks compared to the other tasks. At the same time, the three tasks are dissimilar and present unique challenges from the perspective of text classification. The breast cancer task is a binary classification task to detect self-reported breast cancer patients on Twitter and involves substantial class imbalance as a challenge for NLP; NPMU is a 4-way classification task to classify nonmedical prescription medication use on Twitter (categories include: potential abuse or misuse, non-abuse consumption, drug mention only, and unrelated); and the COVID-19 task is a binary classification task focused on identifying informative COVID-19-related Tweets, which involves a relatively balanced class distribution. For the breast cancer and NPMU classification tasks, we used the same keyword and regular expression filters described in Al-Garadi et al. [41] (i.e., breast cancer-related expressions) and Al-Garadi et al. [42] (i.e., medication names and their spelling variants) to collect additional topic-specific data. For the COVID-19 classification task, we used filtered data from a large dataset from our prior work [56] using the keywords “covid”, “corona virus”, and “coronavirus”. These filters were applied to the PubMed and Twitter datasets, leading to two TSPT datasets for each. Thus, the filtered Twitter data were a topic-specific subset of source-specific data, and the filtered PubMed data were a topic-specific subset of domain-specific data. For comparison, we also created off-topic equivalents of each of these TSPT sets by sampling from the data not detected by the filters from both sources. To summarize, we created 5 pretraining datasets for each classification task (i) topic-specific and domain-specific (from PubMed), (ii) topic-specific and source-specific (from Twitter), (iii) off-topic and domain-specific, (iv) off-topic and source-specific, and (v) generic (i.e., the data from OpenWebText). For fair comparison, we ensured that the off-topic, topic-specific, and generic pretraining sets were of the same sizes for each task: 298,000, 586,000 and 272,000 samples for breast cancer, NPMU and COVID-19, respectively. These sizes were dictated by the number of topic-specific posts we had collected at the time of experimentation. For the web content from OpenWebText and research articles from PubMed, all the documents were chunked into sentences, and each sample is a sentence randomly selected from all the sentences. To further study the effect of pretraining data size for source-specific data, we created three additional large pretraining sets including 1 million samples using the same strategies: (i) topic-specific and source-specific (from Twitter), (ii) off-topic and source-specific, and (iii) generic. PubMed data were not included for these large data experiments due to the availability of the limited topic-specific data related to the three tasks.

#### 2.4.2. Experimental Setup

For MLM, we initialized the models RoBERTa_Base and BERTweet, respectively, and set the learning rate to $4\times 10^{-4}$, the batch size as 4096, and the warm-up ratio as 0.06. The rest of the hyper-parameters were the same as those for pretraining RoBERTa_Base [15]. We trained each model for 100 epochs and used the model from the last checkpoint for fine-tuning. For classification, we performed a limited parameter search with the learning rate $\in \{2\times 10^{-5},3\times 10^{-5}\}$ and fine-tuned each model for 10 epochs. The rest of the hyper-parameters were empirically chosen and are shown in the Appendix A. Because initialization can have a significant impact on convergence in training deep neural networks, we ran each experiment three times with different random initializations. The model that achieved the median performance over the test set was selected to conduct the statistical significance test and report the result.

## 3. Results

### 3.1. Comparison of Pretrained Models

Table 2 presents the performance metrics for the transformer-based models on each task. On most tasks, RoBERTa outperformed BERTweet, and BERTweet only outperformed RoBERTa on 6 out of 11 tasks on tweets. BERTweet performed statistically significantly better than all others on two tasks, and RoBERTa performed statistically significantly better than all others on one task. These results suggest that RoBERTa can effectively capture general text features by pre-training on the general domain data and work well on social media tasks. We also observed that BERT consistently underperformed compared to RoBERTa and BERTweet on all tasks except for SMM4H-21-task6. Although RoBERTa and BERT were both pretrained on general domain data, it is not surprising because RoBERTa was pretrained on a larger dataset with an optimized approach compared to BERT. However, it is interesting to note that BERT consistently underperformed BERTweet although BERTweet was pretrained on smaller data than BERT. These results suggest that pretraining on tweets can reduce the training cost and improve the model efficiency for social media tasks. Although both BERTweet and TwitterBERT were pretrained on Twitter data, the number of tweets used to train TwitterBERT (0.9B tokens) was much smaller than BERTweet (16B tokens), which is likely the reason for the differences in their performances. BioClinical_BERT and BioBERT consistently underperformed on all tasks compared to RoBERTa and BERTweet, and on 19 out of 22 tasks than compared to BERT, despite having undergone DAPT.

### 3.2. Pretraining Results

Table 3 shows the performances obtained on three tasks by models further pretrained on data selected by the different strategies mentioned in the previous section, representing SAPT, DAPT, and TSPT. The table shows that models further pretrained on tweets (SAPT) performed better or comparable to the baseline/off-the-shelf models (RoBERTa_Base and BERTweet), and significantly outperformed the models pretrained on biomedical research papers (DAPT), even with relatively small datasets for extended pretraining. In contrast, there are no statistically significant differences between using the on-topic data and the off-topic data from the same source for the smaller TSPT datasets (i.e., 298K, 586K, and 272K). However, when pretrained using larger datasets (1M), the table shows that the models pretrained on the on-topic data generally obtained better performances than the models pretrained on the off-topic data from the same source, with significantly better performance for the NPMU task. This illustrates that pretraining on data related to the same topic (TSPT) may be effective in some cases. The table also shows that RoBERTa_Base tends to benefit more from SAPT than BERTweet. This may be attributed to the fact that RoBERTa_Base was initially pretrained on generic text while BERTweet was initially pretrained on tweets, and thus RoBERTa_Base can gain more new information from further pretraining on Twitter data compared to BERTweet. The best performance achieved for each of these three tasks is higher than those reported in past literature. We present the implications of these findings in the Section 4.

### 3.3. Document Embedding Transfer Results

Figure 2 visualizes the changes in document embeddings following pretraining and fine-tuning for the three datasets. As we can see, for each type of pretraining dataset, the cosine similarities of the document embeddings before and after pretraining are mostly greater than 0.8, while those of the document embeddings before and after fine-tuning are mostly smaller than 0.6, with a wider spread. This suggests that the embeddings changed substantially after fine-tuning on the classification task compared to the initial pretraining. The same document can be encoded in very different ways depending on what task the model is trained on. The figure also shows that for the breast cancer and COVID-19 tasks, the cosine similarities of the document embeddings before and after pretraining are mostly greater than 0.9. This indicates that the document embeddings hardly changed by pretraining for the breast cancer and COVID-19 tasks. In comparison, for NPMU, the cosine similarities for pretraining show a less concentrated distribution. The large shifts in document embeddings for the NPMU may be one of the reasons for the statistically significant improvement in performance for this task, as depicted in Table 3. The large shifts in document embeddings were also observed in the experiments that obtained statistically significant degradation in performance for the NPMU task. Our work is limited to investigating the relevance between the document embedding shifts and the performance differences. The limitation might be attributed to the cosine similarity measure which requires further investigation in future work.

## 4. Discussion

### 4.1. Performance Comparisons

The consistent high performance of RoBERTa suggests that models pretrained on generic text can still achieve good performance on domain specific social media-based NLP tasks, specifically text classification, and may counterintuitively outperform models pretrained on in-domain (medical) data. The better performance of RoBERTa can be attributed to larger training data, longer training periods and better optimization of hyperparameters. Thus, models pretrained optimally on big, generic text can be a good choice particularly when sufficient domain specific data or computational resources are not available. The relative underperformances of BioClinical_BERT and BioBERT suggest that the effectiveness of DAPT for social media-based health-related text classification tasks can be limited, which may be because of the considerable gap between the languages of the pretraining data and the target tasks (i.e., clinical/biomedical language vs. social media language).

The results in Table 3 illustrate that pretraining on data from the same source (SAPT) and pretraining on data related to the same topic (TSPT) as the target task can be an effective approach for social media-based health-related text classification tasks. However, the effectiveness of SAPT and TSPT differed among three tasks. The most likely possibility for this is that the NPMU task had the most room to improve since the gap between IAA (κ = 0.86) and classifier performance (initial $F_{1}$-score = 0.649) for this task was much bigger than those of the other two (breast cancer: 0.85 vs. 0.892; COVID-19: 0.80 vs. 0.897). Note that IAA in these studies are calculated based on Cohen’s kappa [57], which is not directly comparable with $F_{1}$-scores. We were unable to obtain $F_{1}$-score-based agreements since they were not reported in the original papers describing the annotations. The IAAs, however, do provide estimates about the upper bounds for machine learning performances. While the IAAs are comparable between these studies, the differences in the $F_{1}$-scores clearly indicate the sub-optimal classification performance for the NPMU task. The improvements achieved by TSPT suggest that future researchers may find this strategy to be effective when classification performance is considerably lower compared to IAA.

### 4.2. Embedding Transfer

We investigated the potential reasons for the differences in classification performances by exploring the transfer of the document embeddings during pretraining and fine-tuning. As illustrated in Figure 2, we observed that for breast cancer and COVID-19, the embedding similarities of different models have similar distribution after pretraining on different data, mostly between 0.9 and 1. In comparison, for the NPMU task, the embedding similarities change considerably. This observation may provide a visual explanation for the different performances of the same strategy on different tasks. For the breast cancer and COVID-19 tasks, the document embeddings did not change much after pretraining, indicating that the models may be poorly learning new information. One possible reason for this finding might be that when taking MLM as the training goal, the initial model may be optimal enough to encode the data and may not need extra data. This interpretation is consistent with the pretraining results with larger data in Table 3, which shows that increasing the size of pretraining data does not significantly improve the performance on the breast cancer and COVID-19 tasks, while for the NPMU task, the performance was improved by TSPT with larger data. For the NPMU task, the model representations may have been incomplete and needed more data to improve the representation. Visual analysis, such as the one presented in this paper, may be an effective strategy to decide how much pretraining data is needed for future studies attempting similar supervised text classification tasks.

### 4.3. Implications for Informatics Research

With the rapidly growing inclusion of social media texts for conducting health-related studies, it is imperative to identify NLP strategies that are likely to produce the best results. In most research settings, it is not possible to execute all the different types of pretraining we described in this paper. Furthermore, as reported in recent research, conducting large-scale training/pretraining has associated environmental costs [58,59], and the establishment of effective strategies can significantly lower such costs in future research. Our findings in this paper reveal some simple but effective strategies for improving social media-based health-related text classification tasks. First, large generic models such as RoBERTa and source-specific models such as BERTweet can produce excellent performances in most social media-based text classification tasks. Second, SAPT and TSPT to extend existing pretrained models such as RoBERTa and BERTweet can further improve performance, and they may be particularly useful when existing pretrained models exhibit relative under-performance on a given task. Third, DAPT may not be very effective in improving classification performance for social media tasks, which may have a higher cost-benefit trade-off ratio than SAPT and TSPT. Furthermore, SAPT and TSPT are easy to implement and only require unannotated data. For example, SAPT can be implemented by randomly selecting data from the same source, and TSPT can be implemented by data filtering using topic-related keywords. While our experiments focused solely on text classification tasks, it is likely that these findings will be relevant for other NLP tasks such as information extraction or named entity recognition. It should be noted that the utility of TSPT was only explored for target datasets focused on single topics. Future work should be undertaken to explore the utility of TSPT for target datasets that cover multiple topics.

## 5. Conclusions

We benchmarked the performances of six pretrained transformer-based models on 22 health-related classification tasks involving social media text. We found that RoBERTa outperformed BERTweet on most tasks, and BERTweet consistently outperformed BERT, TwitterBERT, BioClinical_BERT and BioBERT. In addition, we found that pretraining on the data from the same source as the target task (SAPT), in this case social media data, is more effective than pretraining on domain-specific data (DAPT), such as texts retrieved from PubMed. We also found that topic-specific pretraining (TSPT) may in some cases further improve performance, although this strategy may not be as effective as SAPT. Broadly speaking, our experiments suggest that for social media-based classification tasks, it is best to use pretrained models generated from large social media text, and further pretraining on topic-specific data may improve model performances.

## Figures and Tables

**Figure 1 healthcare-10-01478-f001:**
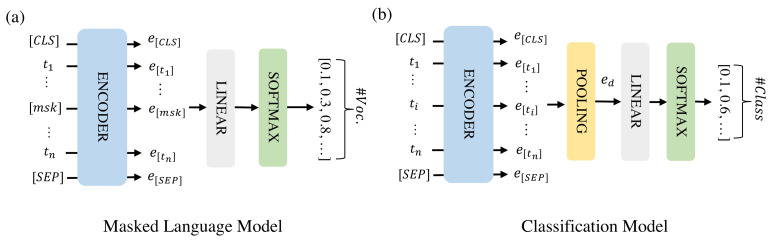
The model architectures for MLM (**a**) and classification (**b**). [CLS] and [SEP] are two special tokens indicating the start and end of the text sequence and [msk] are masked tokens.

**Figure 2 healthcare-10-01478-f002:**
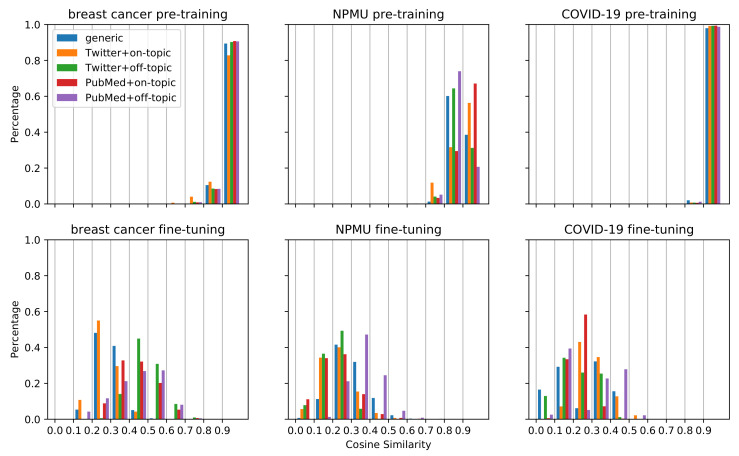
Histograms of the distributions of cosine similarities for the models initialized from RoBERTa_Base and pretrained on 298K, 586K, and 272K samples for the breast cancer, NPMU, and COVID-19 tasks, respectively.

**Table 1 healthcare-10-01478-t001:** Details of the classification Tasks and the data statistics. $P\_F_{1}$ denotes the $F_{1}$-score for the positive class, and $M\_F_{1}$ denotes the micro-averaged $F_{1}$-score among all the classes. * For NPMU, $P\_F_{1}$ denotes the $F_{1}$-score of the non-medical use class. TRN, TST, and L denote the training set size, the test set size, and the number of classes, respectively. EHF denotes the e-health forum. IAA is the inter-annotator agreement, where Task 4 used Fleiss’ K, Task 13 used Krippendorff’s alpha, Task 17–22 provided IAA but did not mention the coefficient they used, and other Tasks used Cohen’s Kappa.

ID	Task	Source	Evaluation Metric	TRN	TST	L	IAA
1	ADR Detection	Twitter	$P\_F_{1}$	4318	1152	2	0.71
2	Breast Cancer	Twitter	$P\_F_{1}$	3513	1204	2	0.85
3	NPMU characterization	Twitter	$P\_F_{1}$ *	11,829	3271	4	0.86
4	WNUT-20-T2 (informative COVID-19 tweet detection)	Twitter	$P\_F_{1}$	6238	1000	2	0.80
5	SMM4H-17-T1 (ADR detection)	Twitter	$P\_F_{1}$	5340	6265	2	0.69
6	SMM4H-17-T2 (medication consumption)	Twitter	$M\_F_{1}$	7291	5929	3	0.88
7	SMM4H-21-T1 (ADR detection)	Twitter	$P\_F_{1}$	15,578	913	2	-
8	SMM4H-21-T3a (regimen change on Twitter)	Twitter	$P\_F_{1}$	5295	1572	2	-
9	SMM4H-21-T3b (regimen change on WebMD)	WebMD	$P\_F_{1}$	9344	1297	2	-
10	SMM4H-21-T4 (adverse pregnancy outcomes)	Twitter	$P\_F_{1}$	4926	973	2	0.90
11	SMM4H-21-T5 (COVID-19 potential case)	Twitter	$P\_F_{1}$	5790	716	2	0.77
12	SMM4H-21-T6 (COVID-19 symptom)	Twitter	$M\_F_{1}$	8188	500	3	-
13	Suicidal Ideation Detection	Reddit	$M\_F_{1}$	1695	553	6	0.88
14	Drug Addiction and Recovery Intervention	Reddit	$M\_F_{1}$	2032	601	5	-
15	eRisk-21-T1 (Signs of Pathological Gambling)	Reddit	$P\_F_{1}$	1511	481	2	-
16	eRisk-21-T2 (Signs of Self-Harm)	Reddit	$P\_F_{1}$	926	284	2	-
17	Sentiment Analysis in EHF (Food Allergy Related)	MedHelp	$M\_F_{1}$	618	191	3	0.75
18	Sentiment Analysis in EHF (Crohn’s Disease Related)	MedHelp	$M\_F_{1}$	1056	317	3	0.72
19	Sentiment Analysis in EHF (Breast Cancer Related)	MedHelp	$M\_F_{1}$	551	161	3	0.75
20	Factuality Classification in EHF (Food Allergy Related)	MedHelp	$M\_F_{1}$	580	159	3	0.73
21	Factuality Classification in EHF (Crohn’s Disease Related)	MedHelp	$M\_F_{1}$	1018	323	3	0.75
22	Factuality Classification in EHF (Breast Cancer Related)	MedHelp	$M\_F_{1}$	524	161	3	0.75

**Table 2 healthcare-10-01478-t002:** Comparison of six pretraining strategies on 22 text classification tasks. The metric for each task is shown along with 95% confidence intervals. The best model for each task is highlighted in boldface. Models that are statistically significantly better than all other models on the same task are underlined.

Task	BERT	RoBERTa	BERTweet	Twitter BERT	BioClinical BERT	BioBERT
ADR Detection	56.3 [48.3–63.6]	60.6 [50.7–64.5]	**64.5** [58.4–70.6]	57.6 [50.6–64.8]	58.9 [51.7–65.3]	60.2 [53.4–66.9]
Breast Cancer	84.7 [81.4–87.7]	**88.5** [85.2–90.3]	87.4 [84.5–90.2]	86.3 [83.3–89.1]	83.0 [79.4–85.8]	83.9 [80.4–86.9]
NPMU	59.5 [55.9–63.0]	61.8 [54.1–61.5]	**64.9** [61.5–68.9]	59.5 [56.0–63.3]	56.8 [53.3–60.6]	52.7 [49.2–56.4]
WNUT-20-T2 (COVID-19)	87.8 [85.7–90.0]	88.7 [87.0–90.9]	**88.8** [86.2–90.9]	87.1 [84.7–89.2]	86.1 [83.9–88.4]	87.4 [85.1–89.6]
SMM4H-17-T1 (ADR detection)	48.6 [44.6–52.8]	**53.4** [47.7–55.5]	50.7 [46.6–54.7]	47.6 [43.3–51.3]	45.5 [41.5–49.1]	44.5 [40.6–48.4]
SMM4H-17-T2 (Medication consumption)	76.8 [75.7–77.8]	79.2 [76.9–79.1]	**79.8** [78.8–80.8]	77.6 [76.6–78.7]	74.7 [73.6–75.7]	75.2 [74.2–76.3]
SMM4H-21-T1 (ADR detection)	68.3 [58.3–77.4]	**71.8** [62.1–80.4]	66.2 [55.7–74.8]	64.9 [53.0–73.9]	64.9 [53.2–73.6]	62.7 [51.0–72.3]
SMM4H-21-T3a (Regimen change on Twitter)	55.5 [48.2–62.7]	**62.1** [55.1–68.8]	57.6 [50.7–64.7]	54.0 [46.4–60.9]	53.6 [46.3–60.6]	55.0 [48.1–61.8]
SMM4H-21-T3b (Regimen change on WebMD)	87.7 [86.1–89.3]	**88.6** [86.9–90.1]	87.6 [85.8–89.2]	87.7 [85.9–89.4]	86.7 [84.8–88.5]	87.1 [85.3–88.9]
SMM4H-21-T4 (Adverse pregnancy outcomes)	86.0 [83.4–88.4]	**89.5** [87.0–91.4]	88.8 [86.4–91.1]	88.4 [86.3–90.7]	83.4 [80.4–86.0]	83.3 [80.4–85.9]
SMM4H-21-T5 (COVID-19 potential case)	69.5 [61.9–75.5]	**75.5** [68.9–81.0]	71.0 [64.6–76.8]	70.9 [64.2–76.8]	65.0 [57.8–71.7]	66.4 [59.0–72.9]
SMM4H-21-T6 (COVID-19 symptom)	**98.4** [97.2–99.4]	98.0 [96.6–99.2]	98.2 [97.0–99.2]	97.8 [96.4–99.0]	97.8 [96.4–99.0]	98.2 [97.0–99.2]
Suicidal Ideation Detection	63.9 [60.0–67.9]	**64.6** [60.4–68.6]	63.3 [59.3–67.3]	59.8 [56.0–64.0]	61.7 [57.4–65.7]	61.7 [57.4–66.1]
Drug Addiction and Recovery Intervention	71.9 [68.2–75.5]	**74.0** [70.4–77.5]	71.9 [68.2–75.2]	69.9 [66.2–73.4]	69.7 [66.2–73.4]	69.7 [66.1–73.2]
eRisk-21-T1 (Signs of Pathological Gambling)	73.9 [57.1–86.2]	**75.0** [59.1–87.7]	67.9 [52.0–81.1]	70.2 [54.5–81.8]	68.1 [50.0–82.1]	62.7 [45.5–76.4]
eRisk-21-T2 (Signs of Self-Harm)	49.1 [32.0–63.8]	**49.3** [34.4–62.9]	48.6 [32.8–61.8]	49.2 [34.0–64.0]	40.0 [25.9–53.3]	45.2 [27.6–60.0]
EHF Sentiment Analysis (Food Allergy)	74.3 [68.1–80.6]	**76.4** [70.2–82.7]	74.3 [68.1–80.6]	71.2 [64.4–77.5]	71.7 [65.4–77.5]	74.9 [68.6-80.6]
EHF Sentiment Analysis (Crohn’s Disease)	77.3 [72.6–81.7]	**79.2** [74.4–83.6]	78.2 [73.5-82.6]	75.4 [70.7–79.8]	75.7 [71.3–80.1]	75.7 [71.0–80.1]
EHF Sentiment Analysis (Breast Cancer)	73.9 [67.1–80.7]	**75.2** [68.3–81.4]	70.8 [63.4–77.6]	72.7 [65.8–79.5]	73.9 [67.1–80.1]	70.2 [62.7–77.6]
EHF Factuality Classification (Food Allergy)	76.1 [69.8-82.4]	**78.0** [71.1–83.6]	76.1 [69.2–82.4]	76.1 [69.2–83.0]	70.4 [62.9–77.4]	76.7 [69.8–83.6]
EHF Factuality Classification (Crohn’s Disease)	83.0 [78.9–87.3]	**85.4** [81.7–89.2]	84.2 [80.2–88.2]	84.8 [81.1–88.5]	82.4 [78.0–86.1]	81.4 [77.1–85.4]
EHF Factuality Classification (Breast Cancer)	75.8 [69.6-82.6]	75.2 [67.7–82.0]	**77.0** [70.2–83.2]	74.5 [67.1–80.7]	75.8 [68.9–82.0]	72.0 [64.6–78.9]

**Table 3 healthcare-10-01478-t003:** Performance metrics obtained by models after pretraining on different data collections. The metric for breast cancer and COVID-19 is the $F_{1}$-score of the positive class, and the metric for NPMU is the $F_{1}$-score for the non-medical use class. RB and BT denote RoBERTa and BERTweet, respectively. Data sizes for extended pretraining are shown at the bottom. The best model for each task is shown in boldface. The models underlined are statistically significantly better than their initial models (i.e., RoBERTa and BERTweet without continual pretraining in Table 2).

Continual Pretraining Data	Initial Model	Breast Cancer		NPMU		COVID-19	
OpenWebText (generic)	RB	87.6 [84.8–90.2]	87.3 [84.4–90.4]	59.5 [55.4–63.1]	57.2 [53.5–61.1]	89.2 [87.1–91.3]	88.5 [86.2-90.6]
	BT	86.5 [83.3–89.2]	87.1 [84.1–89.8]	61.6 [57.8–65.3]	62.1 [58.2-65.2]	88.5 [86.4-90.7]	87.9 [85.8-90.1]
Twitter + off-topic (SAPT)	RB	87.5 [84.5–90.1]	86.4 [83.7–89.2]	65.2 [61.5–68.6]	64.7 [59.0–66.5]	90.8 [88.8–92.6]	89.2 [87.0–91.2]
	BT	86.9 [83.9-89.4]	87.6 [84.7-90.3]	65.7 [62.3-69.0]	64.7 [61.4–67.9]	90.2 [88.0–92.1]	90.1 [88.2–92.1]
Twitter + on-topic (SAPT + TSPT)	RB	**89.7** [87.1–92.0]	88.9 [86.0–91.5]	65.8 [62.5–69.2]	66.0 [63.2–70.0]	90.5 [88.4–92.1]	**91.2** [89.2–92.9]
	BT	89.1 [86.4–91.6]	89.5 [86.9–92.1]	66.7 [63.5–69.9]	**68.0** [64.7–71.4]	90.5 [88.4–92.4]	91.1 [89.1–93.0]
PubMed + off-topic (DAPT)	RB	85.1 [81.9–88.1]	-	55.8 [51.9–59.3]	-	89.0 [87.0–91.2]	-
	BT	85.9 [83.0–88.7]	-	58.8 [55.2–62.1]	-	88.8 [87.0–91.0]	-
PubMed + on-topic (DAPT + TSPT)	RB	85.8 [82.7–88.7]	-	58.6 [55.1–62.4]	-	89.8 [87.7–91.7]	-
	BT	86.9 [84.0–89.5]	-	60.2 [56.6–64.0]	-	89.2 [87.1–91.3]	-
Data size	-	298K	1M	586K	1M	272K	1M

## Data Availability

All datasets used in this study were publicly available at the time. This study did not use or generate any new data. Source code for our model and data preprocessing is available in the Github repository mentioned in the article.

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
