# Peer review of "Comparison of Pretraining Models and Strategies for Health-Related Social Media Text Classification"

_healthcare, 2022, doi:10.3390/healthcare10081478_

Round 1

Reviewer 1 Report

This is a complex and fascinating problem. I have no concerns about the methodology.

However, the project as reported does not report how this research project would help medical services or research teams reliably detect problems and provide solutions to patients.

Texts you analyse about acute health problems include 1, 5 and 7 which are all about ADRs and are all on twitter (minimal text), 13 is on suicidal ideation and 16 is signs of self harm, both on Reddit. 

This research project is not linked to the large and complex medical literatures on ADRs and suicidality and self-harm and no citations are provided and discussed to show the utility and effectiveness of the present research. Please provide detailed searches of these literatures and show how your results assist patients and the health care system.

Reviewer 2 Report

his paper evaluated the effectiveness of different pretrained transformer-based models and strategies. The logic of this article is clear, but there is a slight lack of innovation. There is no innovation in method, only to explore the performance of existing methods on a limited number of data sets. In addition, there are the following specific shortcomings.

(1) Pay attention to the table format. The format of Tables 1-3 is not unified. Whether to use three-line table.

(2) Is there any basis for cosine similarity as a measure of learning ability? Changes in document embeddings before and after pretraining or before and after fine-tuning are not necessarily performance improvements, but may also be performance degradations. Please specify in Sections 2.3.2 and 3.3.

(3) To keep the expression consistent, please use ‘Figure’ instead of ‘Fig.’.

(4) This paper uses the data set provided by the existing literature to verify the performance of the existing models. There is a slight lack of innovation in the method.

(5) Whether using a limited number of data sets can verify the reliability of the conclusion needs further study. It is necessary to consider adding other data sets to improve the reliability of the conclusion.

(6) Pay attention to the format of references. In particular, the format of conference paper references is not unified.

In my opinion, the logical structure of the article is reasonable, but the format needs to be adjusted and the innovation is insufficient. 

Round 2

Reviewer 1 Report

The authors declined to perform a literature review as suggested by the reviewer to show the relevance of their research to actual patients. They state they are demonstrating a technical method to be used by other researchers.  

Reviewer 2 Report

The authors addressed all the comments. I accept the paper in its present form.